# Leaf Carbohydrate Metabolism Variation Caused by Late Planting in Rapeseed (*Brassica napus* L.) at Reproductive Stage

**DOI:** 10.3390/plants11131696

**Published:** 2022-06-27

**Authors:** Yun Ren, Jianfang Zhu, Hui Zhang, Baogang Lin, Pengfei Hao, Shuijin Hua

**Affiliations:** 1Huzhou Agricultural Science and Technology Development Center, Huzhou Academy of Agricultural Sciences, Huzhou 313000, China; yunhuaren@163.com (Y.R.); zjf3700@126.com (J.Z.); 2Zhejiang Agro-Tech Extension and Service Center, Hangzhou 310020, China; zhanghui881007@126.com; 3Institute of Crop and Nuclear Technology Utilization, Zhejiang Academy of Agricultural Sciences, Hangzhou 310021, China; bglin@126.com (B.L.); 11816004@zju.edu.cn (P.H.)

**Keywords:** rapeseed, carbohydrate, enzyme, late planting date, metabolism

## Abstract

Delayed planting date of rapeseed is an important factor affecting seed yield. However, regulation of the leaf carbohydrate metabolism in rapeseed by a late planting date at the reproductive stage is scarcely investigated. A two-year field experiment was conducted to assess the effect of planting dates, including early (15 September), optimal (1 October), late (15 October), and very late (30 October), on leaf growth and carbohydrate biosynthetic and catabolic metabolism at the reproductive stage. The results showed that leaf dry matter decreased linearly on average from 7.48 to 0.62 g plant^−1^ with an early planting date, whereas it increased at first and peaked at 14 days after anthesis (DAA) with other planting dates. Leaf dry matter was the lowest at the very late planting date during the reproductive stage. For leaf chlorophyll content, rapeseed planted at an optimal date maximized at 14 DAA with an average content of 1.51 mg g^−^^1^ fresh weight, whereas it kept high and stable at a very late planting date after 28 DAA. For the carbohydrate catabolic system, acid and neutral invertase (AI and NI, respectively) showed higher activity before 14 DAA, whereas both sucrose synthase (SS) and starch phosphorylase (SP) showed higher activity after 14 DAA. For the carbohydrate biosynthetic system, the activity of sucrose phosphate synthase (SPS) was the highest at the late planting date after 14 DAA, whereas it was at the lowest at the very late planting date. However, the activity of ADP-glucose pyrophosphorylase (AGPase) at the late and very late planting dates was significantly higher than that of the early and optimal plant dates after 21 DAA, which is in accordance with the leaf total soluble sugar content, suggesting that leaf carbohydrate metabolism is governed by a biosynthetic system. The current study provides new insights on leaf carbohydrate metabolism regulation by late planting in rapeseed at the reproductive stage.

## 1. Introduction

Rapeseed (*Brassica napus* L.) is one of the most important oil crops worldwide, which is distributed in Canada, the European Union, China, India, Australia, and others. Almost 80% of winter rapeseed is planted in the Yangtze River, and the remaining spring rapeseed is grown in Northwest China, including Gansu Province, Qinghai Province, Inner Mongolia Autonomous Region, Xinjiang Uyghur Autonomous Region, and Tibet Autonomous Region. Rapeseed is an essential edible oil for humans, who benefit from its high content of ploy-unsaturated fatty acids and healthy fatty acid compositions such as oleic acid and linolenic acid [1,2]. Therefore, increasing seed yield and oil production of rapeseed is a persistent goal for breeders and growers.

Although seeds are the harvesting target in most crops, the yield is closely associated with crop leaves because they are an important organ due to their multiple functions. Firstly, leaves are a producer of photosynthetic products such as carbohydrates, which are essential for plant growth and development [3,4,5]. Secondly, leaf transpiration is one of the driving forces of root water absorption and transportation [6,7,8]. Thirdly, nutrient uptake and accumulation in leaves is necessary for crop life cycling [9,10,11]. Nutrients such as nitrogen assimilates could be absorbed through the roots and transported by the stems and veins to the leaves [12]. Furthermore, leaves can directly absorb nutrients through foliar application [13,14]. Fourthly, leaves are a key factor for plant architecture establishment, including leaf area index, which is highly correlated with crop yield [15,16]. However, the function of crop leaves is not fixed but dynamic. For example, young leaves are considered a sink, whereas adult ones are generally thought of as a strong source [17,18].

The main function of leaves during rapeseed development is similar to that of other crops. However, unlike other crops such as rice and wheat, rapeseed leaves can be divided into three types, which are the long petiole leaves at the young seedling stage, the short petiole leaves or rosette leaves during the late seedling to budding stage, and the sessile leaves at the basal of each branch during the stem elongation and flowering stage [19]. Before floral meristem differentiation initiation, part of the long petiole leaves drops at the basal of the stem. After floral meristem differentiation initiation, the left long petiole and rosette leaves play pivotal roles in carbohydrate and nutrient supply for stem elongating and bud development [20,21]. However, leaves are shaded thoroughly after the middle flowering stage once flowers occupy the canopy. As a result, the remaining long petiole leaves and rosette leaves gradually drop after the end of flowering until the seed-filling stage. At this stage, the nutrients shift from the senescing leaves to other parts, such as the stem and reproductive organs like the bud and developing siliques [12]. Processes including leaf senescing, nutrient transfer from leaves, and leaf falling are affected by many agronomic practices, such as nitrogen application and delayed planting date [22,23].

Late planting always happens with many crops due to reasons such as unsuitable climate [24,25]. Besides climate change, such as continuous rain or drought at the seeding stage, the delay in rice harvesting in the rapeseed–rice rotation system is one of the most important reasons for the late planting of rapeseed. The rapeseed–rice rotation has a long history in the planting system in China, which is mainly adopted by growers in the Yangtze River region. However, in order to further increase the seed yield of rice, the application of long-growth-duration varieties and a change in planting models such as from rice–rice–rapeseed to rice–rapeseed resulted in the extension of rice growth and a delay in rice harvesting. Similar to other crops, delayed planting of rapeseed and other oil crops such as safflower led to a lower seed yield and oil content [26,27]. The yield decrease due to the planting date being linked to the leaf development and physiological status has been widely studied [28,29]. For example, it was found that the leaf emergence rate quickens and the leaf size is reduced with a late planting date in barley [29]. Sinderlar et al. (2010) reported that a leaf area decrease was associated with low yield in corn with a late planting date [30]. The relationship between leaf physiological status and yield change under different planting dates has been also evaluated [31]. However, till now, although many investigations have been performed on the contributions of leaves at the seedling stage on the seed yield of rapeseed [32,33], no reports have been conducted on the correlation between oil yield production and leaf carbohydrate metabolism both with normal and late planting dates in rapeseed at the reproductive stage. As mentioned above, leaves are the major carbohydrate source at the reproductive stage, which can affect seed yield via bud and early silique development. Therefore, we hypothesized that a late planting date would modulate leaf carbohydrate metabolism, which would be an important physiological mechanism of leaf dry matter accumulation and hence a commonly recognized phenomenon of yield reduction.

In our previous study, the results showed that delayed planting significantly influenced leaf carbohydrate profiles [34]. However, the physiological regulation of the carbohydrate metabolism at the reproductive stage under different planting dates in rapeseed has scarcely been reported. In this investigation, we used a widely grown rapeseed variety, Zheyou 50, which is one of the major leading cultivars in the downstream of area of the Yangtze River, China. The main characteristics of the cultivar are high yield (the highest yield gained the Guinness records of Zhejiang Agriculture, which was more than 4500 kg ha^−1^), high oil content (50.1%), and wide adaptability in different growth areas. We first measured the dynamic of leaf dry matter and total sugar content, and then determined the photosynthetic pigment content and carbohydrate metabolic-driven enzymatic activities in rapeseed leaves after initial flowering under different planting dates. The goals of the study were (1) to evaluate the response of leaf dry matter accumulation, chlorophyll, and carbohydrate content to the planting dates during rapeseed reproductive stage; (2) to compare carbohydrate metabolic enzymatic activities under different planting dates; and (3) to elucidate the mechanism on the changed leaf carbohydrate metabolism induced by different planting dates after flowering.

## 2. Materials and Methods

### 2.1. Plant Material and Crop Management

The experiment was conducted during the 2018–2019 and 2019–2020 growing seasons at the experimental station of the Zhejiang Academy of Agricultural Sciences, Hangzhou, China. One rapeseed (*Brassica napus* L.) cultivar, Zheyou 50, was chosen as the plant material. The cultivar Zheyou 50 was bred by the rapeseed breeding team at the Institute of Crop and Nuclear Utilization Technology, Zhejiang Academy of Agricultural Sciences. The soil type in the experimental station is loamy clay (loamy, mixed, and thermic Aeric Endoaquepts). The previous crop was rice grown in a rapeseed–rice rotation system. Urea, calcium superphosphate, potassium oxide, and borax were manually broadcast at the rate of 275, 375, 120, and 15 kg ha^−1^, respectively, as basal fertilizer before sowing. In addition, plants were fertilized by urea as topdressing at the rate of 120 kg ha^−1^ at the end of January in 2019 and 2020. About five rapeseed seeds were directly sown into the soil in a hole at a depth of approximately 3 cm in the plot. The seedlings were thinned to one plant in each hole after one month. The field was not irrigated during the rapeseed growing season, and the mean temperature and precipitation of the two growing seasons is shown in Figure 1.

### 2.2. Experimental Design

The experiment was a completely random block design with three replications. Four planting dates—early (15 September), optimal (1 October), late (15 October), and very late (30 October)—served as the experimental treatment. The recommended optimal planting date was used as a control to compare the effects of different planting dates on leaf carbohydrate metabolism after flowering initiation. The plot was 40 m in length and had eight rows, with spacing between the rows of 0.35 m and between the plants of 0.2 m.

### 2.3. Sampling and Physiological Index Measurement

Plants were harvested from 0 DAA with a 7-day interval. During sampling, six plants were randomly selected in each plot without border ones. Leaves from three plants were mixed and immersed in the liquid nitrogen. The frozen leaf samples were transported to the lab and stored at −80 °C in an ultra-low-temperature refrigerator until they were used for physiological index analysis. Another piece of three plants was immediately taken back to the lab. The leaves were detached from the plants and killed at 90 °C for 0.5 h in an oven. Then the temperature was adjusted to 75 °C to dry the leaf samples until the weight reached a constant value. The samples were cooled in a dryer and weighed by a balance.

### 2.4. Chlorophyll Content Determination

The leaf samples were ground into fine powder with liquid nitrogen and extracted by 800 mL L^−1^ acetone. The total chlorophyll content was determined following the method described by Lichtenthaler [35].

### 2.5. Total Soluble Carbohydrate Content Measurement

The dried leaf samples were ground into fine powder. The extraction and measurement of total soluble carbohydrate content followed the description by Hua et al. [36].

### 2.6. Carbohydrate Metabolism Enzymatic System Assay

Invertase, SUS, and sucrose phosphate synthase (SPS) extraction and determination were performed using the method described by King et al. [37]. Starch phosphorylase (SS) and ADP-glucose pyrophosphorylase (AGPase) activities were measured according to the method of Smith [38], Smith et al. [39], and da Silva [40].

### 2.7. Statistical Analysis

Statistical analysis was performed using SPSS (version 17.0, Chicago, IL, USA). ANOVA was performed on the leaf dry matter, total chlorophyll content, total soluble sugar content, and enzymatic activities. The mean values were compared using Duncan’s test at a probability of 0.05. The results of the statistics are listed in Appendix A.

## 3. Results

### 3.1. Response of Leaf Dry Matter Accumulation to the Planting Date

Rapeseed leaf dry matter showed a decreasing trend at the early planting date (15 September), showing an increasing trend from the day of initial anthesis to 14 DAA and then decreasing under other treatments (Figure 2). The results suggest that leaves on the rapeseed plants at an early planting date would senesce earlier than late planting. Before the leaves dropped off, the rapeseed planted at 1 October showed higher dry matter than other treatments (Figure 2). Leaf dry matter under the treatment of 1 October was above 8 g plant^−1^, whereas it was below 8 g plant^−1^ before 14 DAA under other treatments (Figure 2). Rapeseed leaf growth was seriously inhibited under very late planting date (30 October) because the dry leaf dry matter exhibited the lowest amount at each developing stage except at 14 DAA, which was higher than that sowed at 15 September (Figure 2).

### 3.2. Response of Leaf Total Chlorophyll and Total Carbohydrate Content to the Planting Date

Total chlorophyll content showed an increasing trend from 0 to 14 DAA but very few fluctuations before 21 DAA at optimal and early planting dates (1 October and 15 September, respectively) (Figure 3a and Appendix A). Furthermore, leaf total chlorophyll content under both treatments decreased from 21 DAA very quickly, indicating the fast collapse of chloroplast in the leaf before senescence and drop. The total chlorophyll content at the planting date of 1 October was highest at 14 DAA compared with other planting dates (Figure 3a and Appendix A). When rapeseed plants were planted at late and very late dates (15 October and 30 October, respectively), the total chlorophyll content ranked first and second from 28 DAA on compared with early and optimal planting dates, suggesting the vigorous physiological status of vegetative leaves. The average leaf total chlorophyll content from 28 to 42 DAA for the 30 October and 15 October treatments was 30.4%, 33.3%, 13.1%, and 16.8%, respectively, which is higher than that for the 15 September and 1 October treatments (Figure 3a and Appendix A).

Total carbohydrate content in leaves at early and optimal planting dates showed a decreasing trend from the day of initial anthesis, suggesting that canola plants planted early will correspondingly lead to leaves entering the senescing stage early (Figure 3b and Appendix A). However, when rapeseed plants were under the delayed planting state (15 October), total carbohydrate content in leaves showed a slight increase and then a slight decrease (Figure 3b and Appendix A). The result suggests that the assimilation and dissimilation of the carbohydrates in the leaves was much more equilibrant under the late planting date. However, for the very late planting date (30 October), the total carbohydrate content was low at the beginning of anthesis and 7 DAA and increased from then on. After 28 DAA, the total carbohydrate content remained relatively stable and the highest compared with other treatments (Figure 3b and Appendix A). The result suggests the continuous leaf growth and accumulation of photo-assimilates.

### 3.3. Response of Leaf Carbohydrate Metabolism Enzymatic Activity to Planting Date

#### 3.3.1. Leaf Acid Invertase (AI)

Leaf AI activity under all planting dates decreased sharply from the initial anthesis to 14 DAA (Figure 4a and Appendix A). The leaf AI activity was very low from 14 DAA on, which was reduced by 50% to 75% compared with 0 DAA, and kept rather stable (Figure 4a and Appendix A). The result suggested that the AI in rapeseed leaves mainly functioned at the early reproductive stage (0 to 14 DAA) to catalyze sucrose. As for the influence of planting date on leaf AI activity, leaf AI activity markedly decreased when rapeseed plants were planted under late and very late dates at the early reproductive stage (from 0 to 14 DAA). For example, the average leaf AI activity at the beginning of flowering under the optimal planting date (1 October) was 30% and 38% higher than that at the late and very late planting dates (15 October and 30 October), respectively (Figure 4a and Appendix A).

#### 3.3.2. Leaf Neutral Invertase (NI)

Unlike leaf AI activity, NI activity was weak during the rapeseed reproductive stage (Figure 4b and Appendix A). For the dynamics of leaf NI activity, it increased first from the start of flowering and then decreased for all planting dates except for the very late planting date, which showed a decline from 0 DAA on (Figure 4b and Appendix A). However, a recoverable increment of NI activity was observed at 42 DAA under each planting date (Figure 4b and Appendix A). Generally, leaf NI activity with the very late planting date was the lowest at most rapeseed plant reproductive stages (Figure 4b and Appendix A). Furthermore, leaf NI activity under early and optimal planting dates was significantly higher than that under late and very late planting dates at 42 DAA, which showed 72.3, 82.6, 31.7, and 45.3% increments as compared with late and very late planting dates, respectively (Figure 4b and Appendix A). The results suggests very strong carbohydrate hydrolysis in leaves before dropping.

#### 3.3.3. Leaf Sucrose Phosphate Synthase (SPS)

Generally, leaf SPS activity under each planting date was relatively steady (Figure 4c and Appendix A). For the early planting date (15 September), very high leaf SPS activity was obtained at 0 DAA in 2019 and 7 DAA in 2020, which was 2.4-fold and 2.2-fold higher than that under late and optimal planting dates (15 October and 1 October), respectively (Figure 4c and Appendix A). Leaf SPS activity increased from 0 to 7 DAA (2019)/14 DAA (2020) and then decreased for the optimal planting date (Figure 4c and Appendix A). Notably, rapeseed plants with the late planting date had the highest SPS activity from 21 DAA on in both years, indicating strong requirements of photo-assimilate for delayed leaf development (Figure 4c and Appendix A). However, when rapeseed plants had a very late planting date, the leaf SPS activity was the lowest from 14 to 35 DAA, revealing the weakened sucrose synthetic capacity in the leaves (Figure 4c and Appendix A).

#### 3.3.4. Leaf Sucrose Synthase (SS)

Like leaf AI activity, SS activity was very high under all planting date treatments (Figure 4d and Appendix A). The result clearly demonstrates that carbohydrate catalysis dominated over the carbohydrate metabolism but not synthesis in the leaves during the rapeseed plant reproductive stage. Although the leaf SS activity decreased from the initial flowering to 7 DAA/14 DAA, a rapid increase in leaf SS activity was found till 28 DAA/35 DAA in all planting dates (Figure 4d and Appendix A). At the early reproductive stage (0 to 14 DAA), leaf SS activity was the highest for the very late planting date (Figure 4d and Appendix A). However, from 14 DAA on, leaf SS activity was the strongest for the late planting date, compared with other treatments in 2014 (Figure 4d and Appendix A). The leaf SS activity with the early planting date was the lowest from 14 to 28 DAA as compared with other treatments, indicating the importance of the invertase system for carbohydrate hydrolysis over that of the SS system with an early planting date.

#### 3.3.5. Leaf Starch Phosphorylase (SP)

Leaf SP activity was low and stable from the beginning of flowering to 14 DAA and then increased to different extents under different planting dates (Figure 4e and Appendix A). Generally, the leaf SP activity peaked at 28 DAA for each planting date in both years (Figure 4e and Appendix A). However, at 35 and 42 DAA, leaf SP activity decreased to the lowest activity, which was very weak under all treatments (Figure 4e and Appendix A). At 28 DAA, leaf SP activity for the late planting date (15 October) had the highest value, which was, on average, 36.7, 91.4, and 150.9% higher than that under the optimal (1 October), early (15 September), and very late (30 October) planting dates, respectively (Figure 4e and Appendix A).

#### 3.3.6. Leaf ADP-Glucsoe Pyrophosphorylase (AGPase)

An opposite trend of leaf AGPase activity was found during the rapeseed reproductive stage: The activity in early and optimal planting dates decreased, whereas they increased under late and very late planting dates in both years (Figure 4f and Appendix A). Before 14 DAA, leaf AGPase activity for early and optimal planting dates was significantly higher than that for late and very late planting dates (Figure 4f and Appendix A). However, from 21 DAA on, leaf AGPase activity for late and very late planting dates was significantly higher than that for early and optimal planting dates (Figure 4f and Appendix A). The results suggest that rapeseed leaf under late and very late planting conditions will synthesize starch for leaf growth, whereas starch will be used or degraded in the leaves under early and optimal planting dates.

## 4. Discussion

Although many agronomic practices can influence rapeseed leaf growth, planting date has been paid much attention since it involves simple but very important management. Appropriate planting date can increase crop yield with improved traits such as leaf area [41]. As planting date is delayed, one of the most important influences for rapeseed growth is the loss of heat. In the present study, the average temperature reached 20 °C (Figure 1) for the planting dates from 1 October to 15 October, which means the total effective cumulative temperature would be greatly wasted within 15 days. Therefore, the reduced growth time would deeply affect young rapeseed seedling development and population establishment [42].

Because rapeseed leaves have a long life cycle, which is maintained from true leaf appearance to abscission, the impact of late planting date on leaf development will last the whole duration of leaf growth. Leaf development in rapeseed before budding is mainly in shaping the plant architecture [43]. Once the plants enter the budding stage, another key property during rapeseed development is the simultaneous advancement of vegetative and reproductive growth from the beginning of floral meristem differentiation. At this stage, rapeseed leaves have dual roles: Firstly, they synthesize many assimilates through photosynthesis for new leaf expansion; secondly, the leaves partly act as a source to provide nutrients for rapeseed bud development before the canopy formation. In this context, both functions of rapeseed leaves should be protected for the rapid development of leaves and buds. Normally, rapeseed budding to the end of flowering lasts approximately 40 to 50 days in the downstream area of the Yangtze River region, China. During this process, leaf development is not only influenced by agronomic practices and adverse environmental occurrence, but also its developmental characteristics. For example, as rapeseed stems elongate and the flower layer is established, leaves can be severely shaded, which can significantly affect the efficiency of leaf photosynthesis. In coffee leaves, when plants were shaded in different levels, mean leaf net photosynthesis was kept at a lower level, which decreased by 20% under shade treatments [44]. However, the situation is not absolutely fixed when they suffer from an adverse environment; they can use other methods such as a change in tissue morphology to avoid these adverse encounters [45,46].

When rapeseed has an early planting date, it can accumulate more plant dry matter because of extended growth duration. Our results also explicitly revealed this issue, since the leaf dry matter for the early planting date was lower than that for the optimal planting date and showed a decreasing trend. In addition to the decreasing leaf dry matter, the photosynthetic pigment had nearly no increment at the early reproductive stage, whereas other treatments showed an increasing trend. The observation was in agreement with other reports. For example, an earlier planting date of winter triticale produced more dry matter [47]. On the other hand, another risk is also frequently observed, which is premature senescence, because the rapeseed varieties used in most areas in China belong to a semi-winter type. Soybean plants were planted earlier, leading to the premature flowering, which was highly correlated with temperature and photoperiod [48]. However, as the leaf physiological status alteration, which is the total soluble carbohydrate content in a continuous reduction at an early planting date, kept increasing at late and very late planting dates, it is still unknown how big the influence of light reduction due to the shade by the flower layer is on leaf senescence in rapeseed with different planting dates.

Unlike the early planting date, the reduction in plant growth duration can result in insufficient leaf growth at vegetative and reproductive growth stages with a late planting date because of the lesser leaf dry matter accumulated compared with an optimal planting date (Figure 2). Leaves with insufficient growth condition can further lead to less chlorophyll content at the early reproductive stage (Figure 3). This phenomena clearly uncovers the change in leaf physiological metabolism. Normally, leaf chlorophyll content should be reduced because of both natural senescing and shade stress [49,50]. However, the lesser reduction of leaf chlorophyll content at a late planting date might be due to the compacted plant size, which saves space for reducing the adverse effect of shade stress. Under the same plant density, enough space between rapeseed plants is obviously a benefit for leaf physiological metabolism because of the improvement in irradiation and field humidity [51]. As a result, the highly correlated photosynthetic product, which is carbohydrate content, with chlorophyll content is a reasonable variation trend. Total soluble sugar content in rapeseed leaves is considerably exported to other tissues such as young stems and buds at early and optimal planting dates. However, the carbohydrates in the leaves under late or very late planting dates should be a very small part for export to other tissues because of the slight increase in the sugar content. The results also suggest that catalysis of carbohydrate in rapeseed leaves should not be the dominant metabolism for a late planting date. It was reported that different carbohydrate compositions had various responses to late planting date. For example, fructose content increased by up to 60%, whereas sucrose decreased by up to 50% in cotton under a late planting date treatment [52]. Similarly, in winter wheat, late seeded plants generally deposited more simple sugars but less highly polymerized fructan [53], which was the same trend in rapeseed after 28 DAA under the late planting date treatment.

Rapeseed leaf sucrose catalysis has two independent systems, which are invertase and sucrose synthase [54,55]. Although the function of the two enzymatic systems is the same, they react differently at spatial and temporal levels [56]. In rapeseed leaf, acid invertase at the early stage of anthesis plays a key role in cleaving sucrose into small molecules according to its very strong enzymatic activity at this stage. Furthermore, the higher activity at this stage might be closely correlated with the reallocation of the sugar within different organs such as leaves, buds, and stems. Regardless of the development of the leaves at the early stage of anthesis, large amounts of available cleaved sugars are required for rapeseed bud development. For rapeseed plants, once the suitable developmental condition is ready, buds should appear even if the plant is small, especially at a very late planting date. This is because the rapeseed plants go through vernalization in winter for most semi-winter genotypes [57,58]. Inversely, leaf sucrose synthase showed higher activity at the late flowering stage. Although there is no evidence for the reasons why the sucrose catalytic enzymatic activity kept high throughout the reproductive stage, there are likely some activated factors resulting in the different switches between the two enzymatic systems. Unlike sucrose, very strong starch phosphorylase was found with the late planting date. The continuous leaf expansion for further growth needed a large number of assimilates for the late planting date. This inference was supported by the increased AGPase activity, which is another enzymatic system for starch biosynthesis [59,60]. Unlike the late planting date, the activity of the enzymes was quite different under the optimal planting date. For AGPase activity, it peaked at the early reproductive stage in rapeseed at the optimal planting date, which was an ample sugar source for both rapid leaf and bud development. However, rapeseed leaves also had higher starch phosphorylase activity with the optimal planting date following the late planting date at the late reproductive stage. This should be a benefit for rapeseed stem and silique/seed development. Although rapeseed stems start elongation from budding, the stem is not lignified during this stage and large soluble sugars are utilized by the stem. Furthermore, the stem also acts as a large physiological channel, letting the sugars be transported from leaf to bud. At the middle/late reproductive stage, large amounts of sugars are still required during stem lignification, and those sugars are most likely transported into siliques and seeds. A previous study showed that the canopy acquired the largest capacity and heaviest weight around 30 DAA [34]; therefore, we inferred that the synthesized starch was transported into the sink center, such as developing seeds at the optimal planting date. Under this circumstance, it is reasonable to keep higher enzyme activity both for stem and canopy development because rapid lignification of the stem is quite necessary to enhance its lodging resistance to support the heavy layer of the canopy. Therefore, the stronger enzymatic systems at different reproductive stages were a guarantee for rapeseed’s vigorous vegetative and reproductive growth at the optimal planting date. However, the enzymatic system, including the enzymatic activity and shifting of the expressed stages under different reproductive stages, was significantly changed and limited the plant organ development, which was not benefit for the yield for the late planting date.

## 5. Conclusions

Early planting date, before 1 October, for rapeseed is not recommended, not only for the rice–rapeseed production system for the late harvesting of rice but also due to the adverse effect, mainly referring to the early prematurity and hence the rapeseed yield. Late and very late planting dates resulted in less leaf dry matter and continuous accumulation of chlorophyll and soluble sugar content at the late reproductive stage. The balance between carbohydrate biosynthetic and catabolic systems indicated that the biosynthetic system had the greatest contribution on late planted leaf carbohydrate deposition. The recommendation of planting date of rapeseed was before 15 October in the downstream area of Yangtze River, China. Since the small body of rapeseed led to less leaf dry matter at a late planting date, we strongly suggest that growers increase the planting density, which could be more than 45,000 plants ha^−^^1^, for further study. It should be feasible to compensate the effect of delaying planting date on rapeseed growth by improving plant population under higher planting density to reduce yield losses.

## Figures and Tables

**Figure 1 plants-11-01696-f001:**
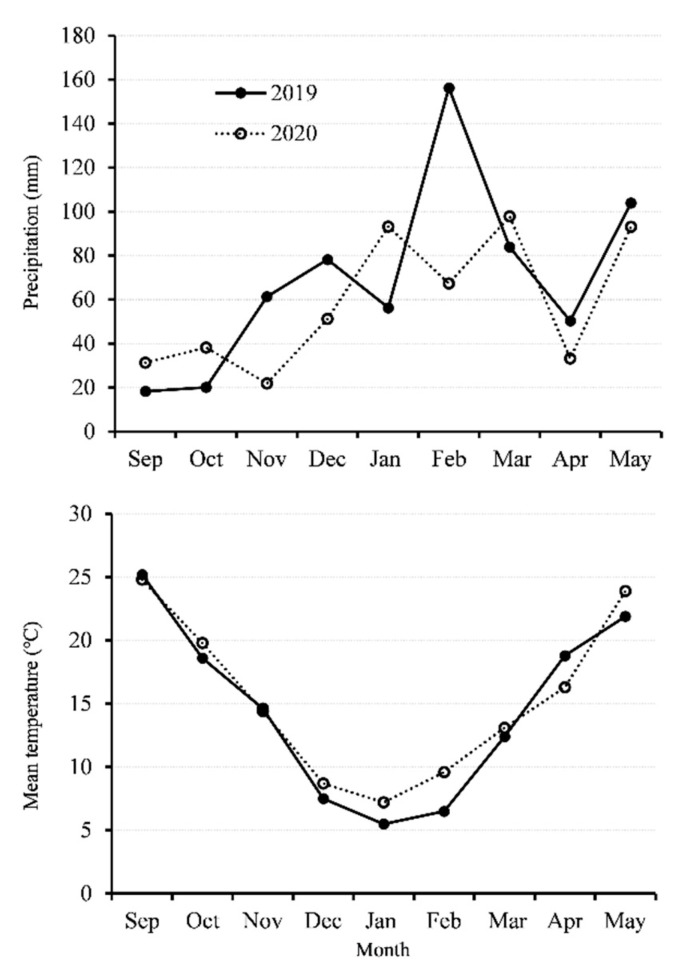
Precipitation and mean temperature during rapeseed growth season in 2018–2019 and 2019–2020.

**Figure 2 plants-11-01696-f002:**
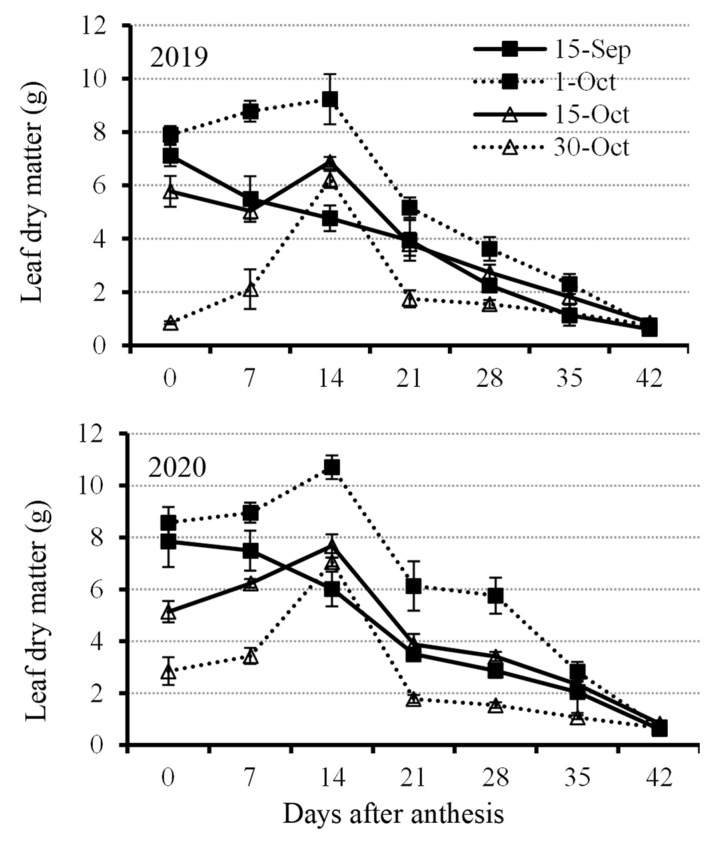
Leaf dry matter accumulation from anthesis with a 7 d interval under four planting dates—early, optimal, late, and very late (15 September, 1 October, 15 October, and 30 October, respectively)—in 2019 and 2020. Bars of each value are standard error.

**Figure 3 plants-11-01696-f003:**
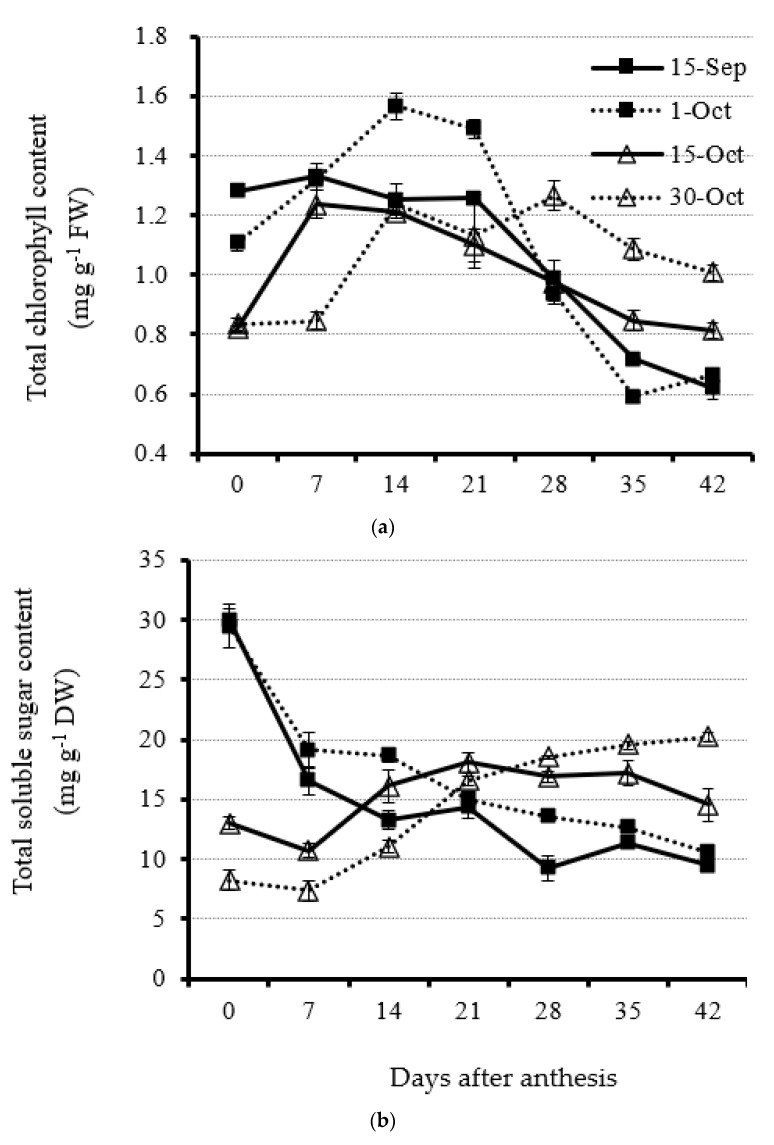
Leaf total chlorophyll (**a**) and total carbohydrate (**b**) content accumulation from anthesis with a 7 d interval under four planting dates—early, optimal, late, and very late (15 September, 1 October, 15 October, and 30 October, respectively)—in 2020. Bars of each value are standard error.

**Figure 4 plants-11-01696-f004:**
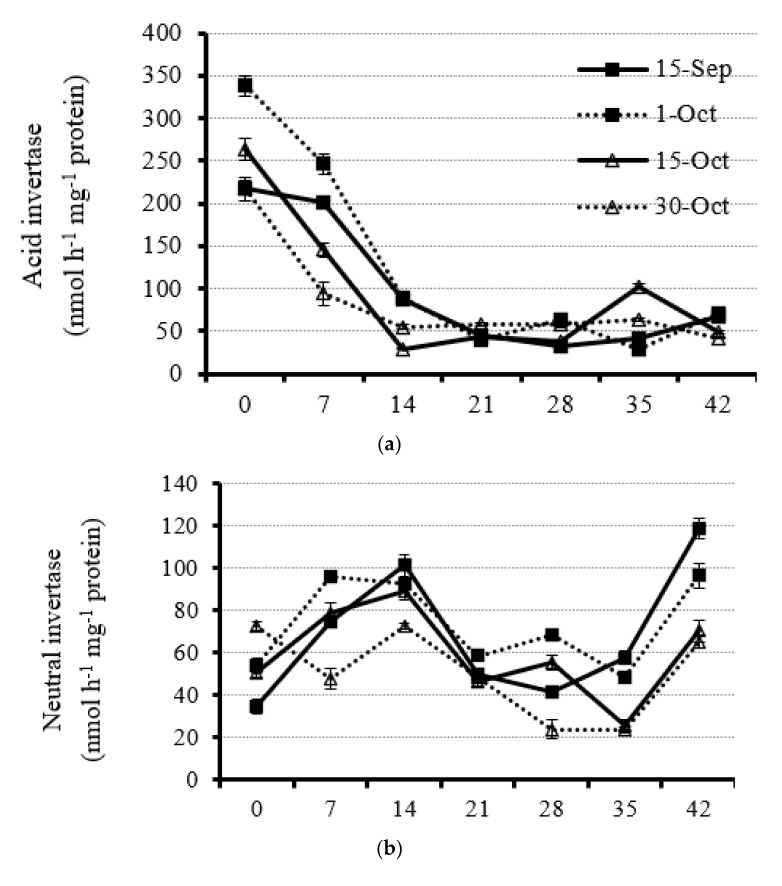
Dynamic of leaf acid invertase (**a**), neutral inverse (**b**), sucrose phosphate synthase (**c**), sucrose synthase (**d**), starch phosphorylase (**e**), and AGPase (**f**) activity from anthesis with a 7 d interval under four planting dates—early, optimal, late, and very late (15 September, 1 October, 15 October, and 30 October, respectively)—in 2020. Bars of each value are standard error.

## Data Availability

Not applicable.

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
