# Peer review of "Leaf Carbohydrate Metabolism Variation Caused by Late Planting in Rapeseed (Brassica napus L.) at Reproductive Stage"

_plants, 2022, doi:10.3390/plants11131696_

Round 1

Reviewer 1 Report

The ms plants-1759576 with the title of Leaf carbohydrate metabolism variation caused by late planting in rapeseed (Brassica napus L.) at reproductive stage assessed the effect of planting date including early (15th Sep), optimal (1st Oct), late (15th Oct), and very late (30th Oct) on leaf growth and photosynthetic metabolism including photosynthetic rate, carbohydrate biosynthesis and catalysis enzymatic systems. The ms has to be improved before it can be accepted in such high quality journal. See my comments below please:

Please do not localize your writing, try to move from local to worldwide in some parts.

L8 avoid using common writing in scientific writing, please revise: becoming more and more

L34-35 please cite this relevant ref. https://doi.org/10.1007/978-981-13-6883-7_19

L49-54 please cite this text with relevant text.

The introduction should be revised and the authors should concentrate also on the importance of rapeseed and rice and link this with their objective because in whole introduction the authors are concentrating on the leaf!

At end of the introduction, authors should add the hypotheses of this work directly after the objectives.

L128 add the missing information for SPSS such as company name and city and country name.

Why the results of second season is better than first season? Can you explain this here and in ms?

Although the results section is well written, but authors have to improve discussion section and make it deeper. The authors have to make it deeper and stronger and show the differences among the different treatment.

Please add an individual section for the conclusion after the discussion section and focus on the most important findings.

 Best regards, 

Reviewer

Reviewer 2 Report

The manuscript gives an interesting insight in the variation of the primary metabolism processes in rapeseed (Brassica napus L.) at reproductive stage due to the late planting due to the rotation agricultural system. Furthermore, the researcher seems very rich in the content and is appropriately designed. I have read it thoroughly and did not find any substantive allegation when it comes to the data presented.

I would suggest it for publication, but some issues needs further explanation:

1) The rationale for the establishment of this work was not clearly founded. The introduction lack the information  of the rapeseed usage which  should put the research in the agricultural context. To the reviewer knowledge rapeseed is cultivated mainly for the canola oil obtained from the seed, when the study is focused on the vegetative organs. It would be valuable for the reader to show how the changes in the primary metabolism are correlated with the production of the oil. What is the reason for the focusing on the primary metabolism only? Determining not only the primary metabolism processes, but also efficiency of the oil production seems justified in the case of the one of the main sources for the global oil production.

2) Line 279. The temperature should be 375 ℃ ?

3) Line 81-84 The extensive information about the cultivar should be moved to the Introduction or Discussion section – the Material and Method section should contain the basic experiments description.

Reviewer 3 Report

After careful reading of the ms I found it suitable for publication, since the authors used suitable and standard methods and analysis. Also discussion is rich, since many important scientific references have been taken into consideration. I suggest to publish the ms after minor revision, since I suggest to improve the ms as follows:

Abstract

 Inroduction

 Insert on which areas rapeseed is grown in the world and China

Materials and Methods

 Discussion

 Strengthen the part of the discussion with 3-4 new references

 Conclusions

- Specify exactly of which sowing density you recommend, or with which densities to perform future experiments in late sowing rapeseed.

References

 - Insert 3-4 more new references

p. 18, line 464: bold the year of printing the reference

 p. 18, line 493: bold the year of printing the reference

 p. 19, line 501: bold the year of printing the reference

Reviewer 4 Report

The Results section in the abstract is unclear and difficult to understand. Lines 15-23 were recommended to rewrite

 The statistical analysis shall be added in Figures 2, 3, 4, 5, 6 7, 8, 9, and 10.

 Combine Figures 3 and 4 into one figure, and select the data of one year in the manuscript. Put the data of the other year in the supplementary information.

 Combine figures 5-10 into one figure, and select the data of one year in the manuscript. Put the data of another year in the supplementary information.

Round 2

Reviewer 1 Report

Dear Editor

The Ms has been improved, but the authors should go through their Ms and correct the trypos errors.

Best regards

Reviewer 

Author Response

Thank you for your constructive suggestions. We have carefully read the Ms and correct the trypos errors.

Reviewer 4 Report

The abstract seemed too long. Please shorten it and improved it.

Author Response

Thank you for your constructive suggestion. We have shortened and improved it.
